# The Role of Hippo Signaling Pathway and ILK in the Pathophysiology of Human Hypertrophic Scars and Keloids: An Immunohistochemical Investigation

**DOI:** 10.3390/cells11213426

**Published:** 2022-10-29

**Authors:** Ilias G. Petrou, Sofia Nikou, Srinivas Madduri, Martha Nifora, Vasiliki Bravou, Daniel F. Kalbermatten

**Affiliations:** 1Department of Plastic, Reconstructive and Aesthetic Surgery, Faculty of Medicine, Geneva University Hospitals and University of Geneva, 1211 Geneva, Switzerland; 2Department of Anatomy, School of Medicine, University of Patras, 26504 Rio Patras, Greece; 3Department of Histopathology, “St.-Andrew” General Hospital of Patras, 26332 Patras, Greece

**Keywords:** keloids, Hippo pathway, ILK, hypertrophic scars, wound healing, scars

## Abstract

Background: Keloids and hypertrophic scars are characterized by abnormal fibroblast activation and proliferation. While their molecular pathogenesis remains unclear, myofibroblasts have been associated with their development. Hippo pathway effectors YAP/TAZ promote cell proliferation and matrix stiffening. Integrin-linked kinase (ILK), a central component of focal adhesions that mediates cell–matrix interactions, has been linked to tissue repair and fibrosis. The aim of this study was to investigate the expression of key Hippo pathway molecules and ILK in hypertrophic scars and keloids. Methods: YAP/TAZ, TEAD4, ILK and a-SMA expression were evaluated by immunohistochemistry in keloids (n = 55), hypertrophic scars (n = 38) and normal skin (n = 14). Results: The expression of YAP/TAZ, TEAD4, ILK and a-SMA was higher in fibroblasts of keloids compared to hypertrophic scars while negative in normal skin. There was a significant positive correlation between the expression of ILK and Hippo pathway effectors. Conclusions: Our results suggest that the deregulation of Hippo signaling and ILK are implicated in keloid and hypertrophic scar formation.

## 1. Introduction

Keloids (KS) and hypertrophic scars (HS) are generated after skin injury that affects both the dermis and epidermis. While their exact pathogenesis remains elusive to date, a prevailing hypothesis is that the deregulation of the molecular mechanisms involved in the normal wound-healing process could lead to skin scarring. The healing process involves highly complex and poorly understood interactions of cells and signaling molecules that lead to an inflammatory response, cell proliferation, deposition of extracellular matrix and tissue remodeling. Any divergence from the normal wound-healing response can result in excess collagen deposition and abnormal extracellular matrix (ECM) remodeling, leading to the formation of KS or HS [1].

Upon wounding, dermal fibroblasts become activated, proliferate, migrate and deposit ECM in order to reconstitute the wound bed. Fibroblast activation results in fibroblast–myofibroblast differentiation. Myofibroblasts are characterized by a contractile apparatus similar to that of smooth muscle and express the marker a-smooth muscle actin (a-SMA) [2]. They hold a central role in wound healing and closure through their capacity to produce strong contractile force and also by virtue of ECM and growth-factor secreting properties. In the late stages of the normal wound-healing process, myofibroblasts are cleared from the wound site. Persistent myofibroblast activation is thought to significantly contribute to skin scarring and the formation of KS or HS [3].

In addition, during the healing process, keratinocytes may undergo epithelial to mesenchymal transition (EMT) and acquire a mesenchymal phenotype, contributing to the formation of KS and HS. A more advanced form of EMT is the epithelial–myofibroblast transition (EMyT), in which epithelial cells acquire a myofibroblast phenotype characterized by the expression of a-SMA [4].

Transforming growth factor-β1 (TGF-β1) exerts a key regulatory role in the wound-healing process as it promotes fibroblast migration and proliferation, increased collagen synthesis and deposition, and re-epithelialization. TGF-β1 is also the major inducer of both EMT and EMyT during the fibrotic process [5]. Furthermore, there is increasing evidence implicating stem cells in the pathogenesis of keloids, as increased numbers of stem cells have been found in keloids [6,7,8]. In addition, stem cells have been shown to produce several mediators, such as histamine, proteases, such as tryptase and chymase, and growth factors, including (PDGF), (VEGF) and (TGF-b1), which promote the fibrotic process [7,9].

The Hippo signaling pathway plays an important role in tissue homeostasis and organ size regulation. It is involved in the complex regulation of tissue-specific stem cells, cell proliferation, cell survival and also tissue healing and regeneration [10]. The core of the Hippo signaling pathway consists of the MST1/2 and LATS1/2 serine/threonine kinases and their corresponding scaffolding proteins SAV1 (Salvador) and MOB1A/B, the transcriptional activators YAP and TAZ and the transcription factors TEAD1-4 [11]. The Hippo pathway is activated when MST1/2 kinase phosphorylates LATS1/2, which, in turn, phosphorylates and inactivates the effectors YAP/TAZ. When the Hippo signaling pathway is inhibited, YAP/TAZ are activated and translocate into the cell nucleus. As YAP and TAZ lack a DNA-binding domain, their access to DNA depends on the association with transcription factors. YAP/TAZ rely mainly on the TEAD family of transcription factors to exert their effects on gene expression [12]. TEAD4 (a transcriptional activator) expression in skin fibroblasts has been shown to promote wound healing, upregulating a number of genes involved in angiogenesis, inflammation and matrix remodeling. The Hippo signaling pathway also interacts with other signaling pathways, including TGF-β [13,14], and previous studies provide evidence of cross-talk between the YAP/TAZ and TGF-β1/Smad signaling pathways during the healing process [15]. Moreover, Hippo signaling has been shown to induce both EMT and EMyT [16,17].

Integrin-linked kinase (ILK), a central signaling and scaffolding focal adhesion protein, is critically involved in integrin-mediated cell–ECM interactions and regulates important cellular functions, including cell proliferation, cell adhesion, migration and EMT [18]. Recent evidence suggests a role of ILK in skin lesions and wound healing [19,20]. Furthermore, there is evidence linking ILK to Hippo signaling as ILK overexpression has been shown to inactivate Hippo signaling by inducing the nuclear accumulation of YAP/TAZ, while inhibition of ILK expression results in activation of the Hippo pathway, with YAP/TAZ phosphorylation and retreat in the cytoplasm and concomitant inhibition of TEAD transcriptional activity [21].

While there is evidence implicating Hippo signaling and ILK in wound healing, their complex role in KS and HS formation is largely unknown. Therefore, this study aims to investigate the expression of the Hippo pathway effectors and ILK as well as their possible interactions in human KS and HS.

## 2. Materials and Methods

### 2.1. Tissue Samples

For the present study, formalin-fixed, paraffin-embedded tissue skin samples from keloids (n = 55), hypertrophic scars (N = 38) and normal skin (n = 14) were retrieved from the archives of the Department of Histopathology of the “St-Andrew” General Hospital of Patras, Patras, Greece. All of the samples have been diagnosed as such by expert pathologists.

### 2.2. Immunohistochemistry

The expression of proteins was studied by immunohistochemistry (IHC) using a standard peroxidase–DAB staining protocol. Briefly, 4-μm-thick paraffin-embedded tissue sections were deparaffinized and rehydrated. For all the antibodies used, tissue sections were subjected to heat-induced epitope antigen retrieval in 0.1 M sodium citrate using an electric microwave oven. Endogenous peroxidase activity was blocked with a 3% H_2_O_2_ solution. Protein blocking was performed with 2% bovine-serum albumin (BSA) in Tris-buffered saline (TBS). The detection of bound primary antibodies was performed using EnVision MiniFLex kit (Agilent, Santa Clara, CA, USA) according to the manufacturer’s instructions. Finally, diaminobenzidine was used as a color substrate and Harris’s hematoxylin as a counterstain. Pictures were taken using a Nikon Eclipse 80i photon microscope and a Nikon DXM 1200c camera with ACT-1C software (Nikon Instruments Inc., Melville, NY, USA). The primary antibodies used are shown in Table 1.

### 2.3. Evaluation of Immunohistochemical Staining

Cytoplasmic and nuclear expression in the epidermis basal layer, blood vessels, fibro/myofibroblasts and skin appendages were evaluated separately. For all proteins, immunoreactivity was graded on a scale of 1–3 according to the intensity of staining and the percentage of positive cells, as already described [22]. Staining intensity was scored as negative (score 0), weak (score 1), moderate (score 2) and strong (score 3). The percentage of positive cells was scored as 0 (staining in less than 1% of cells), 1 (staining in 1–25% of cells), 2 (staining in 26–50% of cells), 3 (staining in 51–75% of cells) and 4 (staining in 76–100% of cells). The two scores were multiplied, resulting in values from 0–12 and the immunoreactivity score was determined as follows: score 1 (multiplication values 0, 1, 2) as low expression, score 2 (multiplication values 3, 4, 6) as medium expression and score 3 (multiplication values 8, 9, 12) as high expression of the studied protein.

### 2.4. Statistical Analysis

The statistical analysis was performed using the statistical package SPSS for Windows, Version 24.0 (IBM SPSS Statistics for Windows, Version 23.0. Armonk, NY, USA: IBM Corp). To test the significance of the differences between the different groups of clinico-pathological parameters in terms of Hippo effectors and ILK, the variables were analyzed by Kruskal–Wallis non-parametric tests (for k independent samples) and Mann–Whitney tests (for two independent samples). The correlation between the expressions of the different proteins was tested by the Spearman Rank Order Correlation test. For all statistical analyses, the values *p* < 0.05 were considered statistically significant.

## 3. Results

### 3.1. Effectors of Hippo Pathway YAP, TAZ and TEAD Are Significantly Overexpressed in Keloids

YAP expression is higher in KS than in HS.

There was no expression of YAP in the control skin. On the contrary, HS and KS showed cytoplasmic and nuclear expression of YAP in the epidermis basal layer (29/38 and 39/55 for cytoplasmic in HS and KS, respectively, and 38/38 and 55/55 for nuclear in HS and KS, respectively) and fibroblasts (36/38 and 36/55 for cytoplasmic in HS and KS, respectively, and 100% of cases for nuclear) while only positive cytoplasmic immunostaining was found in appendages (37/38 and 54/55 for HS and KS, respectively) and blood vessels (37/38 and 55/55 for HS and KS, respectively) as shown in Table 2. There was a statistically significant difference in the expression of YAP in all cell types between controls, HS and KS (*p* < 0.001), and also the expression of YAP in blood vessels and fibroblasts (Figure 1) was significantly higher in KS compared to HS (*p* < 0.001 for blood vessels, *p* < 0.001 for nuclear YAP in fibroblasts and *p* = 0.017 for cytoplasmic YAP in fibroblasts).

TAZ expression is higher in KS compared to HS and control skin.

The expression of TAZ was negative in the appendages, blood vessels and fibroblasts of control skin, while negative or weak cytoplasmic and nuclear expression was observed in the epidermis basal layer. On the contrary, HS and KS (Figure 1) showed cytoplasmic and nuclear expression of TAZ in the epidermis basal layer (28/38 and 39/56 for cytoplasmic in HS and KS, respectively, and 37/38 and 55/55 for nuclear in HS and KS, respectively) and fibroblasts (17/38 and 19/55 for cytoplasmic in HS and KS, respectively, and 37/38 and 54/55 for nuclear in HS and KS, respectively) while only cytoplasmic immunostaining was found in appendages (35/38 and 54/55 for HS and KS, respectively) and blood vessels (36/38 and 54/55 for HS and KS, respectively). There was a statistically significant difference in the expression of TAZ in all cell types between controls, HS and KS (*p* = 0.01 for cytoplasmic TAZ expression in fibroblasts and *p* < 0.001 for expression in all other types). Further, nuclear expression of TAZ in the epidermis basal layer and fibroblasts, as well as the expression of TAZ in appendages and blood vessels, was significantly higher in KS compared to HS (*p* < 0.001), as shown in Table 3.

TEAD4 expression is higher in KS compared to HS and control skin.

TEAD4 expression was negative in all cases of control skin, while in HS and KS, there was nuclear and cytoplasmic expression of TEAD4 in the epidermis basal layer and fibroblasts. (Figure 1) Positive cytoplasmic expression was also observed in appendages and blood vessels. There was a statistically significant difference in the expression of TEAD4 in all cell types between controls, HS and KS (*p* < 0.001). Further, nuclear expression of TEAD4 in fibroblasts, as well as the expression of TEAD4 in blood vessels, was significantly higher in KS compared to HS (*p* = 0.001 and *p* = 0.02, respectively), as shown in Table 4.

There was a significant positive correlation between nuclear YAP, nuclear TAZ and nuclear TEAD4 in the epidermis basal layer (r = 0.573, *p* < 0.001 for nuclear YAP and nuclear TAZ, and r = 0.830 and *p* < 0.001 for nuclear YAP and nuclear TEAD4) and fibroblasts (r = 0.586, *p* < 0.001 for nuclear YAP and nuclear TAZ and r = 0.51 and *p* < 0.001 for nuclear YAP and nuclear TEAD4 and r = 0.511, *p* < 0.001 for nuclear TAZ and nuclear TEAD4). There was also a significant positive correlation between YAP, TAZ and TEAD4 in blood vessels (r = 0.663, *p* < 0.001 for YAP and TAZ, and r = 0.825 and *p* < 0.001 for YAP and TEAD4 and r = 0.582 for TAZ and TEAD4) and in appendages (r = 0.489, *p* < 0.001 for YAP and TAZ, and r = 0.885 and *p* < 0.001 for YAP and TEAD4 and r = 0.452 for TAZ and TEAD4).

### 3.2. a-SMA and ILK Are Significantly Overexpressed in Keloids Compared with Hypertrophic Scars and Skin

Immunostaining for a-SMA in the epidermis basal layer, appendages and fibroblasts showed faint immunopositivity in all cases of normal skin. However, positive a-SMA expression in blood vessel walls was observed in 13/14 (92.9%) control cases, with moderate expression in most of them (11/14, 78%). On the contrary, positive expression of a-SMA was found in the epidermis basal layer in 100% of HS and KS (38/38 and 55/55 cases, respectively), in appendages (36/38 of HS and 52/55 of KS) and blood vessels (37/38 of HS and 54/55 cases of KS). In fibroblasts, we found a relatively strong expression in most cases both for HS and KS (28/38 and 51/55 with scores of more than 2, respectively) (Figure 2). The expression of a-SMA in all cell types was significantly higher in HS and KS compared to control skin (*p* ≤ 0.001). Furthermore, the expression of a-SMA in skin appendages and fibroblasts was significantly higher in KS compared to HS (*p* = 0.039 and *p* = 0.007, respectively), as shown in Table 5. 

The expression of ILK in normal skin was negative, while positive cytoplasmic expression of ILK was observed in the epidermis basal layer in 100% of HS and KS (38/38 and 55/55 cases, respectively), in appendages and blood vessels in 34/38 HS and 52/55 KS, respectively, and fibroblasts in 37/38 and 54/55 cases of HS and KS, respectively (Figure 2). It should be noted that ILK also showed positive nuclear expression in the epidermis basal layer and fibroblasts of HS and KS, as shown in Table 6. There was a statistically significant difference in the expression of ILK in all cell types between controls, HS and KS (*p* ≤ 0.002). Moreover, cytoplasmic expressions of ILK in appendages, blood vessels and fibroblasts were significantly higher in KS compared to HS (*p* = 0.005, *p* = 0.0021 and *p* = 0.017, respectively).

There is also a significant positive correlation between ILK expression and a-SMA in the epidermis basal layer (r = 783, *p* < 0.00100), fibroblasts (r = 0.844, *p* < 0.0010), blood vessels (r = 0.995, *p* < 0.001) and skin appendages (r = 0.986, *p* < 0.001).

### 3.3. YAP/TAZ Activation Correlates with ILK Overexpression in HS and KS

There was a significant positive correlation between the expression of ILK and nuclear YAP, nuclear TAZ and nuclear TEAD4 in fibroblasts (r = 0.629, *p* < 0.001 for ILK and nuclear YAP, r = 0.530 and *p* < 0.001 for ILK and TAZ and r = 595 and *p* < 0.001 for ILK and TEAD4), blood vessels (r = 0.527, *p* < 0.001 for ILK and YAP, r = 0.543 and *p* < 0.001 for ILK and TAZ and r = 432 and *p* < 0.001 for ILK and TEAD4) and in skin appendages (r = 0.407, *p* < 0.001 for ILK and YAP, r = 0.558 and *p* < 0.001 for ILK and TAZ and r = 427 and *p* < 0.001 for ILK and TEAD4) (Figure 3).

## 4. Discussion

Deregulation of the normal wound-healing processes accounts for skin scarring and keloid formation that is characterized by an increased number of fibroblasts and the excessive deposition of extracellular matrix (ECM) in the dermis [23]. In this study, we provide novel evidence implicating Hippo pathway deregulation and ILK overexpression in the formation of keloids and hypertrophic scars.

We showed increased nuclear expression of YAP, TAZ and TEAD in all cell types and mostly fibroblasts of keloids and hypertrophic scars. Moreover, the nuclear expression of all three factors was higher in fibroblasts of keloids compared to HS. In addition, there was a significant positive correlation between nuclear YAP/TAZ and TEAD expression in fibroblasts, suggesting that the activation of YAP and TAZ in synergy with TEAD is implicated in skin scarring. YAP and TAZ are major effectors of the Hippo pathway, as Hippo signaling results in the inactivation of YAP and TAZ by phosphorylation. However, a lack of phospho staining is a limitation of this study and would be considered for further research in the future. Activated YAP and TAZ translocate to the nucleus, where they bind to the transcriptional enhancer factor (TEA)-domain (TEAD) family transcription factors and regulate gene expression to control important cellular functions. In line with our findings, previous evidence suggests that YAP/TAZ are essential drivers of epidermal stem cell and fibroblast proliferation during skin development and repair [24]. YAP/TAZ has been reported to be implicated in skin wound healing by modulating the expression of the transforming growth factor (TGF)-β1 signaling pathway [10]. Nuclear YAP/TAZ acts as a cofactor for Smad signaling and enhances Smad nuclear localization that, in turn, promotes the TGF-β signaling [25].

Furthermore, previous studies have shown that the downregulation of YAP/TAZ RNA by applying small interfering RNA (siRNA) to mouse skin wound sites leads to delayed healing. The knockdown of either YAP or TAZ RNA reduced TGF-b1 expression at the wound site. The expression of CTGF, Smad-2, p21 and Smad-7 in the wound area was also altered in YAP siRNA- and TAZ siRNA-treated wounds. These results suggest that YAP and TAZ regulate the wound-healing process by modulating TGF-b1 signaling [10]. Moreover, it has been previously demonstrated that the inhibition of YAP/TAZ and TEAD4 via either gene knockdown or verteporfin treatment, a recently identified YAP inhibitor, significantly inhibits fibroblast proliferation, induces fibroblast apoptosis and downregulates COL1A1 production by keloid fibroblasts. Verteporfin was first identified as a YAP inhibitor that mainly blocks the interaction between YAP and TEAD and also inhibits the expression of YAP/TAZ [10]. Based on the above, YAP/TAZ implication in keloid formation, as evidenced in the present study, may have clinical implications as inhibitors of YAP may be exploited in the future for therapeutic purposes.

We also showed that ILK is overexpressed in fibroblasts of keloids and hypertrophic scars with significantly higher expression in keloids compared to hypertrophic scars and control skin, providing evidence that implicates ILK in skin scarring. In accordance with our findings, previous studies reported that ILK may be involved in the regeneration of various tissues after injury [26]. It has been demonstrated that ILK mediates pathological fibrosis by mediating TGF-β-induced EMT in diverse cell types, such as mammary epithelial cells and renal tubular epithelial cells. ILK knockout significantly interrupted TGF-β-induced EMT as well as α-SMA expression [18]. ILK also plays an important role in the EMT process induced by TGFβ-1 in mammary epithelial cells, and the ILK/Rictor complex has been proposed as a potential molecular target for preventing/reversing EMT [27]. In further support of our findings, ILK has been shown to mediate TGF-β1-induced a-SMA expression and myofibroblast differentiation in dermal fibroblasts, and the overexpression of ILK in isolated stellate cells led to enhanced motility and adhesion as well as increases in a-SMA and type-I collagen expression [26,28]. These data suggest that ILK is implicated in myofibroblast activation and skin scarring.

Importantly, we here showed that the expression of Hippo pathway transcriptional activators (YAP, TAZ and TEAD4), as well as a-SMA, are significantly correlated with ILK expression, providing novel evidence linking Hippo and ILK signaling in skin scaring. In accordance, a link between ILK and Hippo pathway has been previously evidenced in cancer, as the inhibition of ILK in different types of cancer resulted in the activation of the the Hippo pathway components MST1 and LATS1 with concomitant inactivation of YAP/TAZ transcriptional co-activators and TEAD-mediated transcription [21].

## 5. Conclusions

In conclusion, our study showed that nuclear YAP, TAZ and TEAD expression, a-SMA positive myofibroblasts and ILK expression are significantly higher in keloids compared with hypertrophic scar tissue and control skin, suggesting their involvement in the pathophysiology of skin scarring. Moreover, a significant positive correlation between the activation of YAP, TAZ and TEAD and the overexpression of ILK has been evidenced in keloids and hypertrophic scars, suggesting an interplay between Hippo and ILK pathways in the process of aberrant wound healing. Further studies, however, are needed to elucidate the exact mechanisms of interaction between Hippo and ILK pathway as well as downstream target genes of activated YAP and TAZ during skin scarring. It would also be helpful to correlate the expression of Hippo pathway regulators and interactors with other clinico-pathological features, such as the anatomical location of HS/KS, pre-existing wounds and other medical conditions that affect wound healing. Overall, future studies elucidating the role of the Hippo pathway and ILK in skin wound healing and scarring may lead to the development of novel therapeutic options for keloids.

## Figures and Tables

**Figure 1 cells-11-03426-f001:**
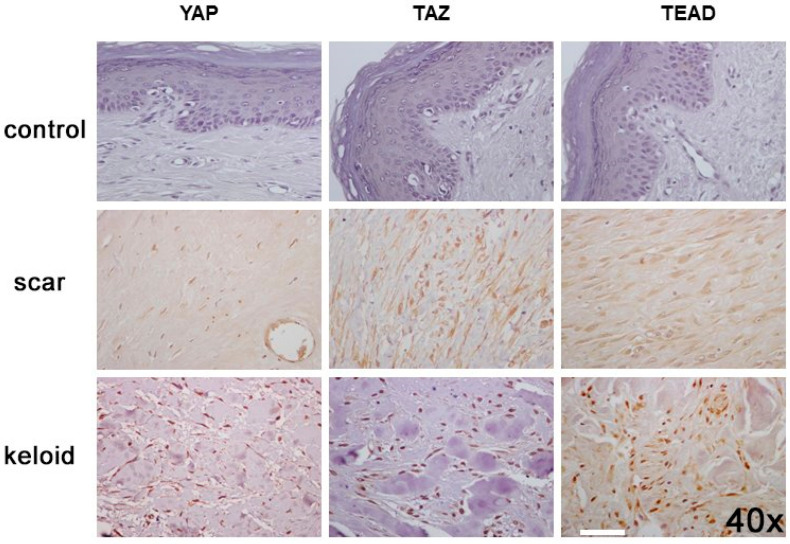
Hippo pathway effectors YAP, TAZ and TEAD4 are overexpressed in hypertrophic scars and keloids. Representative cases of control skin (**upper panel**), hypertrophic scars (**middle panel**) and keloids (**lower panel**) showed stronger immunopositivity in fibroblasts for YAP, TAZ and TEAD4 in keloids compared to scars (magnification 40×). Scale bar set at 20 μm (ImageJfiji software).

**Figure 2 cells-11-03426-f002:**
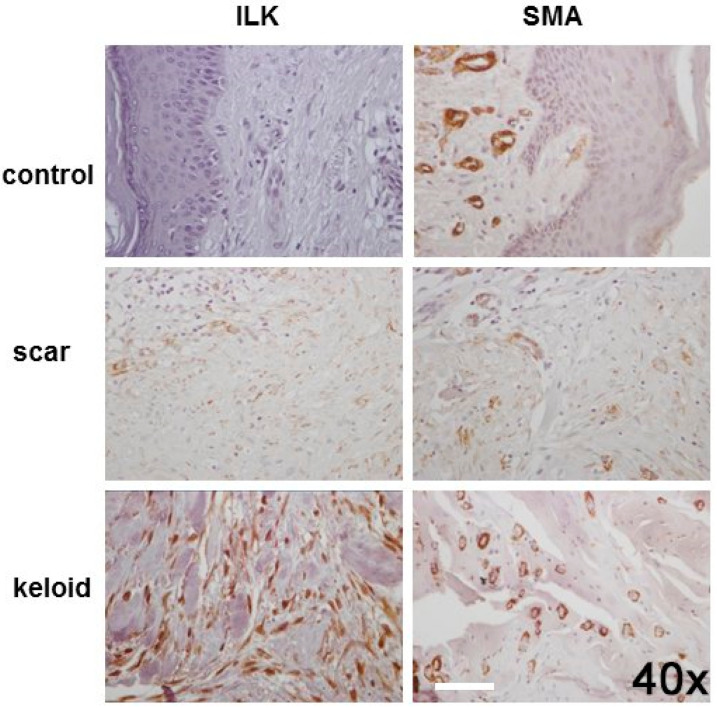
ILK and a-SMA are overexpressed in hypertrophic scars and keloids. Representative cases of control skin (**upper panel**), hypertrophic scars (**middle panel**) and keloids (**lower panel**) show stronger immunopositivity for ILK and a-SMA in keloids compared to scars (magnification 40×). Scale bar at 20 μm (ImageJfiji software).

**Figure 3 cells-11-03426-f003:**
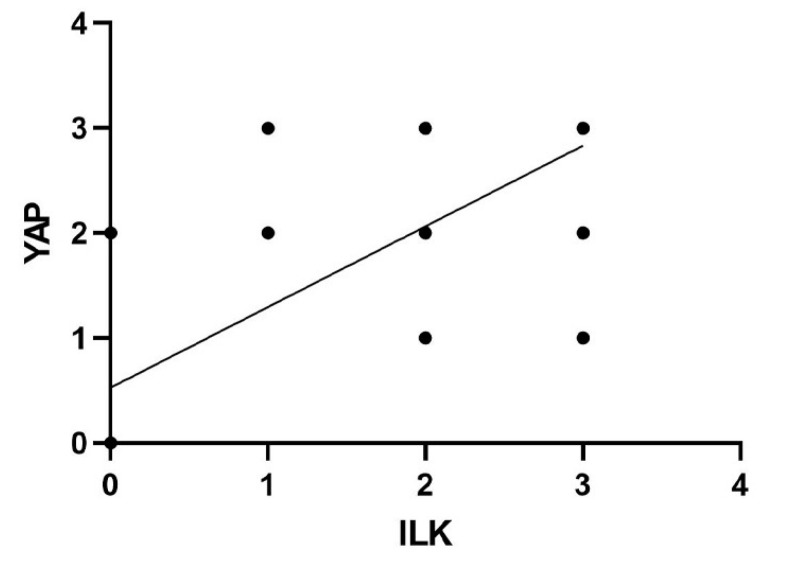
Plots depicting the correlation between ILK and expression of YAP, TAZ and TEAD4 in fibroblasts.

**Table 1 cells-11-03426-t001:** Antibodies used in immunohistochemistry.

	Isotype Class	Dilution	Catalog Number	Company
YAP	Mouse monoclonal	1:300	#1A12	Cell Signaling Technology (CST) (Danvers, MA, USA)
TAZ	Rabbit polyclonal	1:60	(H-70) sc-48805	Santa Cruz Biotechnology (Santa Cruz, CA, USA)
Tead4	Rabbit polyclonal	1:50	#PA5-21977	Thermo Fisher Scientific (Waltham, MA, USA)
a-SMA	Rabbit polyclonal	1:800	#PA5-117765	Thermo Fisher Scientific (Waltham, MA, USA)
ILK	Mouse monoclonal	1:50	(65.1) sc-20019	Santa Cruz Biotechnology (Santa Cruz, CA, USA)

**Table 2 cells-11-03426-t002:** Immunohistochemical analysis of YAP expression in control skin, hypertrophic scars and keloids.

		Control (N = 14)	Hypertrophic Scars (N = 38)	Keloids (N = 55)		
	Score ^a^	N (%)	N (%)	N (%)	*p*-Value ^b^	*p*-Value ^c^
YAP cytoplasmic expression						
Epidermis Basal Layer	0	14 (100)	9 (23.7)	16 (29.1)	<0.001	0.446
	1	0(0)	28 (73.7)	39 (70.9)		
	2	0(0)	1 (2.6)	0 (0)		
	3	0 (0)	0 (0)	0 (0)		
Appendages	0	14 (100)	1 (2.6%)	1 (1.8)	<0.001	0.274
	1	0 (0)	1 (2.6%)	1 (1.8)		
	2	0 (0)	23 (60.5)	28 (50.9)		
	3	0 (0)	13 (34.2)	25 (45.5)		
Blood vessels	0	14 (100)	4 (10.3)	3 (5.5)	<0.001	<0.001
	1	0 (0)	1 (2.6%)	0 (0)		
	2	0 (0)	19 (50.0)	7 (12.7)		
	3	0 (0)	17 (43.6)	48 (87.3)		
Fibroblasts	0	14 (100)	2 (5.3)	19 (34.5)	<0.001	0.017
	1	0 (0)	33 (86.8)	30 (54.5)		
	2	0 (0)	3 (7.9)	6 (10.9)		
	3	0 (0)	0 (0)	0 (0)		
YAP nuclear expression						
Epidermis Basal Layer	0	14 (100)	0 (0)	0 (0)	<0.001	0.270
	1	0 (0)	0 (0)	0 (0)		
	2	0 (0)	16 (42.1)	17 (30.9)		
	3	0 (0)	22 (57.9)	38(69.1)		
Fibroblasts	0	14 (100)	0 (0)	0 (0)	<0.001	<0.001
	1	0 (0)	2 (5.3)	0 (0)		
	2	0 (0)	21 (55.3)	3 (5.5)		
	3	0 (0)	15 (39.5)	52 (94.5)		

^a^ scoring of immunohistochemical expression was performed as described in the materials and methods. ^b^ Kruskal–Wallis: differences in immunohistochemical expression between control, scars and keloids. ^c^ Mann–Whitney: differences in immunohistochemical expression between scars and keloids.

**Table 3 cells-11-03426-t003:** Immunohistochemical analysis of TAZ expression in control skin, hypertrophic scars and keloids.

		Control (N = 14)	Hypertrophic Scars (N = 38)	Keloids (N = 55)		
	Score ^a^	N (%)	N (%)	N (%)	*p*-Value ^b^	*p*-Value ^c^
TAZ cytoplasmic expression						
Epidermis Basal Layer	0	7 (50)	10 (26.3)	16 (29.1)	0.093	0.184
	1	7 (50)	22 (57.9)	38 (69.1)		
	2	0 (0)	6 (15.8)	1 (1.8)		
	3	0 (0)	0 (0)	0 (0)		
Appendages	0	14 (100)	3 (7.9%)	2 (3.6)	<0.001	<0.001
	1	0 (0)	8 (21.1%)	1 (1.8)		
	2	0 (0)	24 (63.2)	10 (18.2)		
	3	0 (0)	3 (7.9)	42 (76.4)		
Blood vessels	0	14 (100)	2 (5.3)	2 (3.6)	<0.001	<0.001
	1	0 (0)	3 (7.9%)	0 (0)		
	2	0 (0)	16 (42.1)	5 (9.1)		
	3	0 (0)	17 (44.7)	48 (87.3)		
Fibroblasts	0	14 (100)	21 (55.3)	37 (67.2)	0.010	0.255
	1	0 (0)	15 (39.5)	17 (31.0)		
	2	0 (0)	2 (5.3)	1 (1.8)		
	3	0 (0)	0 (0)	0 (0)		
TAZ nuclear expression						
Epidermis Basal Layer	0	7(50.0)	1 (2.6)	2 (3.6)	<0.001	<0.001
	1	7 (50.0)	1 (2.6)	0 (0)		
	2	0.0%	21 (55.3)	11 (20.1)		
	3	0 (0)	15 (39.5)	42 (76.3)		
Fibroblasts	0	14 (100)	2 (5.3%)	4 (7.2)	<0.001	<0.001
	1	0 (0)	1 (2.6)	0 (0)		
	2	0 (0)	20 (52.6)	6 (11.0)		
	3	0 (0)	15 (39.5)	45 (81.8)		

^a^ scoring of an immunohistochemical expression was performed as described in the materials and methods. ^b^ Kruskal–Wallis: differences in immunohistochemical expression between control, hypertrophic scars and keloids. ^c^ Mann–Whitney: differences in immunohistochemical expression between hypertrophic scars and keloids.

**Table 4 cells-11-03426-t004:** Immunohistochemical analysis of TEAD4 expression in control skin, hypertrophic scars and keloids.

		Control (N = 14)	Hypertrophic Scars (N = 38)	Keloids (N = 55)		
	Score ^a^	N (%)	N (%)	N (%)	*p*-Value ^b^	*p*-Value ^c^
TEAD4 cytoplasmic expression						
Epidermis Basal Layer	0	14 (100)	26 (68.4)	42 (76.4)	0.199	0.398
	1	0 (0)	12 (31.6)	13 (23.6)		
	2	0 (0)	0 (0)	0 (0)		
	3	0 (0)	0 (0)	0 (0)		
Appendages	0	14 (100)	0 (0)	1 (1.8)	<0.001	0.636
	1	0 (0)	0 (0)	1 (1.8)		
	2	0 (0)	27 (71.1)	33 (60)		
	3	0 (0)	11 (28.9)	20 (36.4)		
Blood Vessels	0	14 (100)	0 (0)	0 (0)	<0.001	0.02
	1	0 (0)	0 (0)	0 (0)		
	2	0 (0)	17 (44.7)	12 (21.8)		
	3	0 (0)	21 (55.3)	43 (78.2)		
Fibroblasts	0	14 (100)	31 (81.6)	48 (87.3)	0.219	0.453
	1	0 (0)	7 (18.4)	7 (12.7)		
	2	0 (0)	0 (0)	0 (0)		
	3	0 (0)	0 (0)	0 (0)		
TEAD4 nuclear expression						
Epidermis Basal Layer	0	14 (100)	0 (0)	0 (0)	<0.001	0.902
	1	0 (0)	0 (0)	0 (0)		
	2	0 (0)	8 (78.9)	11 (20)		
	3	0 (0)	30 (57.9)	44 (80)		
Fibroblasts	0	14 (100)	0 (0)	0 (0)	<0.001	0.001
	1	0 (0)	2 (5.3)	0 (0)		
	2	0 (0)	16 (42.1)	9 (16.4)		
	3	0 (0)	20 (52.6)	46 (83.6)		

^a^ scoring of immunohistochemical expression was performed as described in the materials and methods. ^b^ Kruskal–Wallis: differences in immunohistochemical expression between control, hypertrophic scars and keloids. ^c^ Mann–Whitney: differences in immunohistochemical expression between hypertrophic scars and keloids.

**Table 5 cells-11-03426-t005:** Immunohistochemical analysis of a-SMA expression in control skin, hypertrophic scars and keloids.

		Control (N = 14)	Hypertrophic Scars (N = 38)	Keloids (N = 55)		
a-SMA Expression	Score ^a^	N (%)	N (%)	N (%)	*p*-Value ^b^	*p*-Value ^c^
Epidermis Basal Layer	0	14 (100)	0 (0)	0 (0)	0.001	0.077
	1	0 (0)	0 (0)	0 (0)		
	2	0 (0)	4 (10.5)	15 (28.3)		
	3	0 (0)	34 (89.5)	38 (71.7)		
Appendages	0	14 (100)	2 (5.2)	3 (5.5)	<0.001	0.039
	1	0 (0)	8 (21.0)	4 (7.3)		
	2	0 (0)	23 (60.5)	34 (61.8)		
	3	0 (0)	5 (13.3)	14 (25.5)		
Fibroblasts	0	14 (100)	3 (8.0)	3 (5.5)	<0.001	0.007
	1	0 (0)	7 (18.4)	1 (1.8)		
	2	0 (0)	11 (28.9)	13 (23.6)		
	3	0 (0)	17 (44.7)	38 (69.1)		
Blood vessels	0	1 (7.1)	1 (2.6)	1 (1.8)	<0.001	0.189
	1	0 (0)	1 (2.6)	1 (1.8)		
	2	11 (78.6)	8 (21.2)	6 (10.9)		
	3	2 (14.3)	28 (73.6)	47 (85.5)		

^a^ scoring of immunohistochemical expression was performed as described in the materials and methods. ^b^ Kruskal–Wallis: differences in immunohistochemical expression between control, hypertrophic scars, and keloids. ^c^ Mann–Whitney: differences in immunohistochemical expression between hypertrophic scars and keloids.

**Table 6 cells-11-03426-t006:** Immunohistochemical analysis of ILK expression in control skin, hypertrophic scars and keloids.

		Control (N = 14)	Scars (N = 38)	Keloids (N = 55)		
	Score ^a^	N (%)	N (%)	N (%)	*p*-Value ^b^	*p*-Value ^c^
ILK cytoplasmic expression						
Epidermis Basal Layer	0	14 (100)	0 (0)	0 (0)	0.002	0.305
	1	0 (0)	0 (0)	2 (3.6)		
	2	0 (0)	8 (21.0)	11 (20.1)		
	3	0 (0)	30 (79.0)	42 (76.3)		
Appendages	0	14 (100)	4 (10.5)	3 (5.5)	<0.001	0.025
	1	0 (0)	8 (21.0)	4 (7.3)		
	2	0 (0)	21 (55.3)	34 (61.8)		
	3	0 (0)	5 (13.2)	14 (25.5)		
Blood vessels	0	14 (100)	4 (10.5)	3 (5.5)	<0.001	0.005
	1	0 (0)	8 (21.0)	1 (1.8)		
	2	0 (0)	10 (26.4)	13 (23.6)		
	3	0 (0)	16 (42.1)	38 (69.1)		
Fibroblasts	0	14 (100)	1 (2.6)	1 (1.8)	<0.001	0.017
	1	0 (0)	2 (5.2)	1 (1.8)		
	2	0 (0)	9 (23.7)	5 (9.1)		
	3	0 (0)	26 (68.5)	48 (87.3)		
ILK nuclear expression						
Epidermis Basal Layer	0	14 (100)	12 (31.5)	24 (43.6)	<0.001	0.138
	1	0 (0)	20 (52.6%)	27 (49.1)		
	2	0 (0)	6 (15.9)	3 (5.5)		
	3	0 (0)	0 (0)	1 (1.8)		
Fibroblasts	0	14 (100)	16 (42.2)	27 (49.1)	0.001	0.430
	1	0 (0)	19 (50.0)	24 (43.6)		
	2	0 (0)	1 (2.6)	0 (0)		
	3	0 (0)	2 (5.2)	4 (7.3)		

^a^ scoring of immunohistochemical expression was performed as described in the materials and methods. ^b^ Kruskal–Wallis: differences in immunohistochemical expression between control, hypertrophic scars and keloids. ^c^ Mann–Whitney: differences in immunohistochemical expression between hypertrophic scars and keloids.

## Data Availability

Not applicable.

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
