# Peer review of "The Role of Hippo Signaling Pathway and ILK in the Pathophysiology of Human Hypertrophic Scars and Keloids: An Immunohistochemical Investigation"

_cells, 2022, doi:10.3390/cells11213426_

Round 1
Reviewer 1 Report
In the manuscript entitled “The role of Hippo signaling pathway and ILK in the pathophysiology of human hypertrophic scars and keloids: An immunohistochemical investigation”, Ilias G. Petrou et al. demonstrated that YAP/TAZ, TEAD4, ILK and a-SMA were overexpressed in human fibroblasts of keloids, compared to hypertrophic scars while the normal skin was negative.
This paper does not add new pieces of relevant information to the current knowledge.Moreover, the paper is not well written and results are poorly reported.
1. Since phosphorylation of YAP/TAZ recruits downstream proteins that stimulate cytoplasmic retention or proteolytic degradation. Phosphorylated of YAP/TAZ, not the total amount of YAP and TAZ should be investigated.
2. For correlative analysis, the immunoexpression levels should be presented in a plot.
3. The quality of repesentaive images from control group stained with SMA is awful.
4. Extra spaces should be deleted, like:“he YAP / TAZ and TGF-β1 / Smad ”, “is the epithelial – myofibroblast transition”
5. Spaces should be added in “and re-epithelialization.Studies”, “process.[13]A more advance”.
Tons of mistakes are found in this manuscript.
6. Details on the imaging analysis process used for the calculation of the percentage of positive cells should be added.
Reviewer 2 Report
1. Figure 1 and 2 lack scale bars; please add them. The text in the image is of low resolution and will appear blurry when zoomed in; please increase the resolution. In addition, it is recommended to add a title to each panel on the left side of Figure 1, just like Figure 2.
2. In the “Materials and Methods” section, please provide the dilution of each antibody.
3. In line 221, “a scoring of a a” is confusing, please correct it.
4. In line 75, there is a missing space before “Inhibition of ILK”. In line 82, there is a missing space before “A more advanced form”. Please correct them.
5. In line 113, “H2O2” please change to “H2O2”.
6. There are extra spaces in the following positions, please correct them, such as, line 114, 175, 177, 180, 263, 368 and 379.
7. In line 368, there is a missing full stop before “Overexpression of ILK”.
Round 2
Reviewer 1 Report
I am satisfied with the author's reply.
Reviewer 2 Report
The manuscript has been carefully revised and is recommended for acceptance.